# Evaluation of an Adapted Semi-Automated DNA Extraction for Human Salivary Shotgun Metagenomics

**DOI:** 10.3390/biom13101505

**Published:** 2023-10-11

**Authors:** Victoria Meslier, Elisa Menozzi, Aymeric David, Christian Morabito, Sara Lucas Del Pozo, Alexandre Famechon, Janet North, Benoit Quinquis, Sofia Koletsi, Jane Macnaughtan, Roxana Mezabrovschi, S. Dusko Ehrlich, Anthony H. V. Schapira, Mathieu Almeida

**Affiliations:** 1MetaGenoPolis, INRAE, Université Paris-Saclay, 78350 Jouy-en-Josas, Francechristian.morabito@inrae.fr (C.M.);; 2Aligning Science Across Parkinson’s (ASAP) Collaborative Research Network, Chevy Chase, MD 20815, USAs.koletsi@ucl.ac.uk (S.K.); r.mezabrovschi@ucl.ac.uk (R.M.); s.ehrlich@ucl.ac.uk (S.D.E.); 3Department of Clinical and Movement Neurosciences, UCL Queen Square Institute of Neurology, University College London (UCL), London WC1E 6BT, UK; 4Research Department of Hematology, Cancer Institute, University College London (UCL), London WC1E 6BT, UK; 5Liver Failure Group, Institute for Liver and Digestive Health, University College London, London WC1E 6BT, UK

**Keywords:** DNA extraction, automation, oral microbiome, shotgun metagenomics

## Abstract

Recent attention has highlighted the importance of oral microbiota in human health and disease, e.g., in Parkinson’s disease, notably using shotgun metagenomics. One key aspect for efficient shotgun metagenomic analysis relies on optimal microbial sampling and DNA extraction, generally implementing commercial solutions developed to improve sample collection and preservation, and provide high DNA quality and quantity for downstream analysis. As metagenomic studies are today performed on a large number of samples, the next evolution to increase study throughput is with DNA extraction automation. In this study, we proposed a semi-automated DNA extraction protocol for human salivary samples collected with a commercial kit, and compared the outcomes with the DNA extraction recommended by the manufacturer. While similar DNA yields were observed between the protocols, our semi-automated DNA protocol generated significantly higher DNA fragment sizes. Moreover, we showed that the oral microbiome composition was equivalent between DNA extraction methods, even at the species level. This study demonstrates that our semi-automated protocol is suitable for shotgun metagenomic analysis, while allowing for improved sample treatment logistics with reduced technical variability and without compromising the structure of the oral microbiome.

## 1. Introduction

In recent years, several studies have highlighted the detection and potential influence of the oral microbiome in numerous human diseases such as liver cirrhosis, type 2 diabetes, rheumatoid arthritis, COVID-19 infection, colorectal and pancreatic cancers [1,2,3,4,5,6,7,8,9,10], and neurodegenerative diseases such as Parkinson’s disease (PD) [11,12,13,14]. A convenient way to analyze the oral microbiome is to study saliva, as it does not require an invasive collection procedure compared to gingival, tongue, dental plaque or cheek microbiome samples [15,16]. The saliva is nonetheless a complex ecosystem to study, as it is one of the most diverse human microbial ecosystems with fecal microbiota, with diverse microorganisms detected such as bacteria, fungus, and viruses [17,18,19]. Yet, one of the challenges of studying the oral microbiome is the high rate of human DNA that could represent up to 90% of the sequenced DNA when performing shotgun metagenomic sequencing [20]. For these reasons, analyzing the saliva microbiome using shotgun metagenomics requires a higher sequencing depth than the fecal microbiome (which usually contains less than 1% of human DNA in healthy individuals) to obtain equivalent microbiome taxonomical and functional depth analysis [20].

Thus, to explore this ecosystem efficiently, at both taxonomical and functional levels, various sampling strategies exist, including commercial kits, some of which are adapted for microbial sample collection, preservation, and exploration through shotgun metagenomic analysis [21]. Some methods are known to be efficient but could represent a risk for the technician’s health and are difficult to automate, such as the method relying on phenol-chloroform. Indeed, these methods require the careful separation of the aqueous and organic phases after centrifugation to segregate the remaining phenol chloroform, as these toxic reagents could impair downstream applications on the microbial DNA [22]. As a result, both traditional alcoholic DNA salts precipitation protocols and silica-column-based kits were selected by the International Human Microbiome Standards (IHMS) consortium as reliable, standardized procedures to optimize DNA yield and quality for whole shotgun metagenomic analysis [22].

The next evolution in increasing the throughput for DNA extraction is automation, usually relying on adapting existing kits to high-throughput machines. These robots, such as the QIAsymphony SP instrument from Qiagen, the KingFisher Flex instrument from Thermo Fisher, or the Maxwell instrument from Promega, can automate the extraction and purification of DNA from human samples. However, these robots require specific sample logistics depending on the treated sample type, that needs to be evaluated compared to manual gold standard processes. In this paper, we propose an adapted semi-automated protocol following IHMS recommendations for the shotgun metagenomic analysis (Protocol P1) [23] of saliva samples collected with DNA Genotek’s OMNIgene•ORAL (OM-501), in contrast to DNA Genotek’s recommended DNA extraction protocol for such samples based on the QIAamp PowerFecal Pro kit (Protocol P2). The aim of this study is to assess the potential bias generated by the semi-automated Protocol P1 and validate its use to improve sample treatment logistics, without compromising the composition of the oral microbiome.

## 2. Materials and Methods

### 2.1. Ethics Declaration

Participants were recruited through the Remote Assessment of Parkinsonism Supporting Ongoing Development of Interventions in Gaucher Disease study (RAPSODI) (http://rapsodistudy.com, accessed on 2 October 2023) and via the PD FRONTLINE platform [24]. All participants who consent to take part in the RAPSODI or PD FRONTLINE studies are screened for genetic variants in the GBA1 and LRRK2 genes. As part of their participation in the RAPSODI or PD FRONTLINE study, participants may be asked to provide a saliva sample for analysis of their oral microbiome, which is part of this project. Ethical approval was provided by the London—Queen Square Research Ethics Committee (REC reference: 15/LO/1155).

### 2.2. Salivary Collection Using OMNIgene ORAL OM 501 Device

All participants were asked to collect one saliva sample using the OMNIgene•ORAL|OM-501 sample collection kit (DNAGenotek, Ottawa, ON, Canada). Participants were provided with written instructions on how to collect the sample and asked not to eat, drink, smoke or chew gum for 30 min before collection [25]. Samples were collected at the time of the in-person clinical assessment at the Royal Free Hospital (London, UK) and stored at −80 °C until further steps. Patients with Parkinson’s disease and their partners or family members were invited to participate.

### 2.3. Overall Strategy

Before DNA extraction from the salivary sampling device, a pre-process was performed by heating the whole OMNIgene Oral OM 501 tubes into an incubator at 50 °C for 2 h (Figure 1). Then, we split the samples as follows: (i) four raw samples were assigned to each of the DNA extraction protocols P1 or P2 to evaluate each protocol’s efficiency in extracting microbial DNA by accounting for individual’s inter-variability, and (ii) three pairs of samples were randomly chosen and mixed to produce three new samples (referred to hereinafter as pooled samples) to evaluate the differences between protocols (Appendix A). For each raw or pooled sample, a volume between 1.5 and 2 mL was then transferred into a 2 mL Sarstedt tube, depending on the amount of saliva produced by each donor. A total of seven samples (4 raw and 3 pooled) were processed for each of the two DNA extraction protocols, described in the sections below. After DNA extraction, whole genome shotgun (WGS) sequencing was performed using Ion Proton technology followed by bioinformatics and biostatistics analysis.

### 2.4. DNA Extraction Procedures

The two DNA extraction procedures performed in this study included both mechanical and chemical lysis steps, and differed regarding the purification procedures (semi-automated purification for Procedure 1 and manual for Procedure 2), as presented in the sections below and in Figure 2, following the manufacturers recommendations.


**Procedure 1: MGP Standard Operating Procedure**


DNA extraction was performed on an aliquot of salivary samples following the MGP SOP001 v1 (https://mgps.eu/standard-operating-procedure, accessed on 2 October 2023), adapted from the IHMS (International Human Microbiome Standards) SOP 07 V2 protocol (http://www.human-microbiome.org/index.php?id=Sop&num=005, accessed on 2 October 2023) (Figure 2). The protocol is fully available at https://www.protocols.io/view/protocol-for-dna-extraction-from-saliva-samples-us-dm6gpjm11gzp/v1 (accessed on 2 October 2023) [23]. First, 250 µL of Guanidine Thiocyanate (4 M; Sigma G6639) and 40 µL of N-Lauroyl Sarcosine (10% *w*/*v*; Sigma L9150) were added to 500 µL of saliva sample, and gently mixed before adding 500 µL of N-Lauroyl Sarcosine (5% *w*/*v*) and incubating the tubes in a dry incubator for 1h at 70 °C. A mechanical lysis was performed by adding 750 µL of autoclaved glass beads (0.1 mm) to the tubes that were mixed for 10 min at 25 s^−1^ speed using a bead beater (MixerMill MM400, Retsch, Éragny, France). Then, 15 mg of PVPP (Sigma 77627) was added and the tubes were vigorously shaken before a centrifugation at 18,200× *g*, 4 °C for 5 min. The supernatant was recovered in a sterile 2 mL tube and 500 µL of TENP solution (50 mM Tris pH 8, 20 mM EDTA pH 8, 100 mM NaCl and 1% PVPP) was added to the remaining pellet, mixed well, and centrifuged at 18,200× *g*, 4 °C for 5 min. The supernatant was recovered and pooled to the previous one and centrifuged for 10 min at 18,200× *g*, 4 °C. Lysis extracts were then purified using the QIAsymphony SP instrument (Qiagen, Venlo, The Netherlands), following the QIAsymphony DSP Virus/Pathogen midi kit version 1, that included automated DNA particles binding to magnetic beads, washing and elution steps. Purified DNA samples were re-suspended in 110 µL of the elution buffer (AVE) provided in the kit. DNA extracts were stored at −20 °C before further steps.


**Procedure 2: QIAGEN QIAamp Power Fecal Pro**


We used the QIAamp PowerFecal Pro kit as recommended by the manufacturer’s instructions (available at https://www.qiagen.com/ (accessed on 2 October 2023)) (Figure 2). Briefly, 250 µL of saliva sample was added in a dry bead tube with 750 µL of PowerBead Solution and 60 µL of solution C1. The tubes were heated for 65 °C for 10 min and mechanical lysis was performed using a Vortex Adapter at maximum speed for 10 min. Different series of chemical lysis were performed using C2 and C3 solutions, before proceeding to the purification using C4 and C5 solutions and the MB Spin Column (refer to the kit protocol for additional details). The elution was performed with 110 µL of C6 solution provided in the kit. DNA extracts were stored at −20 °C before further steps.

### 2.5. DNA Yield and Quality Evaluation

DNA quantification was performed on each tube using the DNA-binding fluorescent Quant-it dsDNA BR assay (Thermo Fisher Scientific, Waltham, MA, USA) following the manufacturer’s instructions. DNA quality was determined using the Genomic 50 kb kit by capillary electrophoresis using the Fragment Analyzer (Agilent Technologies Inc., Santa Clara, CA, USA). Three DNA fractions were determined depending on the distribution of the size of the DNA: fraction 1, fragments ≤ 1000 bp; fraction 2, >1000 bp and ≤10,000 bp; and fraction 3 > 10,000 bp. The areas under the fragment analyzer curves were determined for each fraction 1 to 3 for each sample and reported as a percentage of the total area under the curve to finely estimate the overall DNA quality.

### 2.6. Shotgun Metagenomics

The full description for library construction and shotgun sequencing using Ion Torrent technology was previously described [26]. Briefly, libraries were built with 1 µg of high-molecular-weight DNA and using the Ion Plus Fragment Library kit (Thermo Fisher Scientific) to a targeted length of 150 nt. After purification using magnetic beads, libraries were quantified and quality checked using the HS Small fragment kit (Fragment Analyzer, Agilent), with an expected length of 240 nt and expected quantity above 1000 pM. Ion Proton and Ion GeneStudio S5 sequencers were used to generate 20 million of 150nt raw reads per sample, as described before [26].

### 2.7. Bioinformatics: Read Mapping and Microbial Species

Raw reads obtained from Ion Torrent sequencing technology were quality trimmed using AlienTrimmer v2.0 (http://ftp://ftp.pasteur.fr/pub/gensoft/projects/AlienTrimmer/ (accessed on 2 October 2023)) [27] with the following parameters: “-k 10 -l 45 -m 5 -p 40 -q 20” and human-related reads were removed using Bowtie2 v2.5.1 (http://bowtie-bio.sourceforge.net/bowtie2/index.shtml (accessed on 2 October 2023)) [28] using the Human genome reference GRCh38 (accession GCA_000001405.15, nucleotide identity > 90%). High quality reads were mapped onto the 8.4 million gene oral catalog (https://entrepot.recherche.data.gouv.fr/dataset.xhtml?persistentId=doi:10.15454/WQ4UTV (accessed on 2 October 2023)) [29] using the METEOR software v1 (https://forgemia.inra.fr/metagenopolis/meteor (accessed on 2 October 2023)), using a two-step procedure as described before [30]. The resulting gene count table was then downsized to 1 million mapped reads using the R package MetaOMineR v1.31 (https://github.com/eprifti/momr (accessed on 2 October 2023) or https://forgemia.inra.fr/metagenopolis/momr (accessed on 2 October 2023)) to take into account differences in sequencing depth; this was normalized for gene length and finally transformed into a frequency matrix using the FPKM strategy. The 8.4 million gene oral catalog was previously organized, using a MSPminer tool, into 853 MSP species [29,31,32], corresponding to clusters of co-abundant genes used as proxies for microbial species, and containing core and accessory genes. Taxonomical annotation was assigned using the Genome Taxonomy Database [33]. MSP definition and taxonomy are available from Data INRAe (https://entrepot.recherche.data.gouv.fr/dataset.xhtml?persistentId=doi:10.15454/WQ4UTV (accessed on 2 October 2023)) [29]. MSP relative abundance was defined as the mean abundance of 100 marker genes corresponding to core genes with the highest correlations altogether. MSP species with less than 10% of their marker genes detected were considered absent (their abundance was set to 0). Relative abundances at higher taxonomic ranks were computed as the sum of the MSP abundance that belong to a given taxon.

### 2.8. Biostatistics

Statistical analysis was carried out using the R software v3.6.0 (http://www.r-project.org/ (accessed on 2 October 2023)) [34]. Comparisons between two groups were performed using unpaired Wilcoxon ranked tests, and FDR was controlled by *p*-values correction using the Benjamini–Hochberg (BH) procedure. Results were declared significantly different for a *p*-value less than 0.05. Correlations between variables were determined using Spearman’s correlations. Alpha and beta diversity indices were determined by using the vegan R package v2.5.7 (http://cran.r-project.org/web/packages/vegan/index.html (accessed on 2 October 2023)) [35]. PCA analysis was performed on a log10 transformed msp count table using FactoMineR v2.4 and Factoextra v1.0.5 packages (http://factominer.free.fr/index.html and https://cran.r-project.org/web/packages/factoextra/index.html (accessed on 2 October 2023)) [36,37]. Box plots were computed using the ggpubr R package v0.2.1 (https://CRAN.R-project.org/package=ggpubr (accessed on 2 October 2023)) [38], heatmaps were created with the ComplexHeatmap v2.2.0 and pheatmap package v1.0.12 [39,40] (https://www.rdocumentation.org/packages/pheatmap/versions/0.2/topics/pheatmap (accessed on 2 October 2023)), and metagenomic analyses were performed using the momr R package v1.31 (https://forgemia.inra.fr/metagenopolis/momr (accessed on 2 October 2023)).

## 3. Results

### 3.1. Overview of the Analytical Workflow

The impact of DNA extraction procedures was evaluated on salivary samples collected from individuals of the UK-based RAPSODI/PD Frontline cohort and conducted using whole shotgun metagenomic sequencing (Figure 1 and Appendix A). The two DNA isolation protocols both comprised chemical and mechanical lysis steps but differed in terms of purification strategies (magnetic beads and silica membrane for P1 and P2 protocols, respectively) and automation (semi-automated vs manual treatment for P1 and P2 protocols, respectively). Two types of samples were processed: (i) raw samples, corresponding to original salivary samples, and (ii) pooled samples, obtained after mixing pairs of raw samples and redistributing them equitably between the two protocols before proceeding to shotgun metagenomic sequencing and analysis (Figure 1).

### 3.2. Evaluation of Standard DNA Extraction Quality Parameters

#### 3.2.1. DNA Concentration and Fragment Size

We first evaluated the impact of metagenomic DNA extraction procedures on technical parameters, including DNA quantity and quality (Figure 3, Appendix A). We found that DNA quantity was not influenced by the protocol (Figure 3A; *p*-value = 0.62) nor by sample type, although higher variability was observed between the raw and pooled samples obtained from the P2 protocol (Appendix A; *p*-values > 0.4). However, we observed a significant difference in the size of the genomic peak (*p*-value = 0.007, Figure 3B), that corresponds to the sequenced DNA fraction and used to estimate the overall DNA quality (Appendix A). The P1 protocol permitted the conservation of a higher DNA fragment size (mean = 56 ± 4.5 kb) compared to the P2 protocol (mean = 43.1 ± 9 kb; Figure 3B and Appendix A), and significantly prevented the degradation of the genomic DNA. Indeed, DNA fragments of sizes below 10 kp, corresponding to the degradation of the DNA that should have been used for shotgun sequencing, was significantly enriched in the P2 protocol (fraction 2; Appendix A).

#### 3.2.2. Metagenomic Read Quality and Mapping

When considering metagenomic sequencing read quality, we observed no differences between protocols for the total number of sequenced reads and filtered human-related reads (Appendix A), nor for the number of high quality reads (Figure 3C) and for microbial-related reads after mapping to a representative human oral microbial gene catalog (Figure 3D).

### 3.3. Impact of DNA Isolation Procedures on the Oral Microbiome

#### 3.3.1. Alpha Diversity Indices

We measured variations in alpha diversity metrics by evaluating the microbial gene richness and Microbial Species Pangenomes (or MSP) richness, corresponding to microbial gene clusters of different strains from same species reconstructed by binning procedures using oral metagenomic samples [29,31,32]. We did not find significant differences for gene and MSP richness considering the pooled samples (Figure 4A, Appendix A), nor considering all samples (Appendix A). Although not significant, we nonetheless observed a positive trend between microbial richness with the genomic peak (Appendix A).

#### 3.3.2. Structure and Composition of the Oral Microbiome

We also evaluated the effects of DNA extractions on the overall composition of the oral microbiome by performing Spearman correlations based on gene composition (Figure 4B) and MSP composition (Appendix A). We found that the oral microbial composition was similar between the two DNA extraction procedures. PCA and Bray-Curtis dissimilarity indices analyses on the MSP composition further confirmed this finding (Figure 4C and Appendix A). Likewise, we found few variations between protocols on the microbial composition at the phylum, family, and genus taxonomic ranks (Figure 5A–C and Appendix A), nor variations for MSP (trends for two differentially abundant species between pooled samples; *p*-values Wilcoxon test < 0.05, Appendix A).

## 4. Discussion

In this paper, we adapted a manual saliva DNA extraction protocol for semi-automation processing and followed the IHMS standards to improve sample treatment logistics and quality for shotgun metagenomic analysis. We compared this new procedure to the recommended manual one. As the number of saliva samples collected is limited, and to reduce the effort of collecting saliva samples for the donors, our benchmark was based on two types of samples: (i) raw saliva samples coming from a single donor to reflect the restricted amount of raw material collected [41] with strong but various human DNA concentration [20,42], and (ii) pooled samples from different donors to evaluate each protocol’s efficiency using identical samples.

It is difficult to anticipate whether a semi-automated procedure could be as effective as a manual DNA extraction procedure, as this could depend on the intrinsic properties of the samples such as the overall microbial composition [43]. As observed in our study using manual extraction only, the amount of microbial DNA obtained from the various donors’ raw saliva was highly variable, a finding that is consistent with previous studies [42,44]. This variability was also observed on the retrieved host DNA, representing up to 87% of the total sequenced DNA [20]. Even in this highly variable microbial and human DNA setting, our proposed semi-automated DNA extraction P1 protocol was efficient in preserving DNA fragment size and quality without significantly affecting the oral microbial composition compared to the recommended manual protocol [44]. This was expected as the P1 protocol shares the lineage of the IHMS protocols, which were carefully selected among various kits and manual protocols for their reliability, reproducibility, and for the quality of the obtained DNA. Indeed, it relies on magnetic particles technology, which usually provides larger DNA fragments than column-based protocols [22]. Thus, the advantage of this semi-automated process is to preserve high-quality DNA while facilitating the logistics of sample processing, as the operator can benefit from machine running times to perform other experiments [45]. Indeed, we found that this semi-automated protocol involves four times less human-operating time than the corresponding manual method (IHMS SOP 07, https://human-microbiome.org/index.php (accessed on 2 October 2023)). However, this does not prove that the methods used are effective to extract DNA for all types of microorganisms, as the initial microbial composition is unknown. It would be interesting to benchmark these methods (manual and automated) on mock saliva samples with known initial composition that reflect the natural oral microbial complexity and human-to-microbiota cells proportion, which is still unavailable at the time of our study.

We also anticipate that our protocol could improve metagenomic analysis relying on long read metagenomic sequencing due to the longer generated DNA fragment, impacting the metagenomic assembly, functional annotation, and binning strategies [46]. Future development could consider reducing the initial sample volume to unlock multi-protocol benchmarking or multi-omics study, but this will require new steps in the automation process to reduce possible well-to-well contamination when analyzing a low biomass sample such as saliva [47]. Moreover, as this protocol derives from a stool-based protocol, also referred to as the Godon method, which is itself based on bacterial DNA isolation from a fluidized-bed reactor fed by vinasses [48], we speculate that it can be used to isolate high-quality DNA from a large variety of samples.

Our experience shows that, despite requiring an initial investment to buy the machines necessary for automation or semi-automation, these strategies generally lead to a decrease in the cost for DNA extraction. This is because human operating time is more expensive than consumables. However, it is only relevant as part of a high-throughput pipeline because processing only a handful of samples in a semi-automated pipeline is most likely to generate high extra costs. In this context, semi-automation involving a single automation platform and various manual operations, as well as automation with low-cost liquid handlers, are a good compromise between high-throughput and flexibility at reasonable costs [49].

Finally, this study is based on passive salivation oral samples, considered as a gold standard in oral microbial analysis, because (i) it allows the collection of a large quantity of saliva, (ii) it contains the microbes in transit into the digestive tract that could participate to human gut dysbiosis, and (iii) it does not require invasive tools for sample collection [16]. However, passive drool collection is not practical for all patients, such as those suffering from dry mouth (xerostomia) or the advanced stages of neurodegenerative diseases. It would be interesting to validate our protocol on other oral sites, such as the tongue dorsum, for which a sufficient amount of material can be sampled, and therefore be efficiently processed through whole shotgun metagenomic analysis [16,18,50].

## Figures and Tables

**Figure 1 biomolecules-13-01505-f001:**
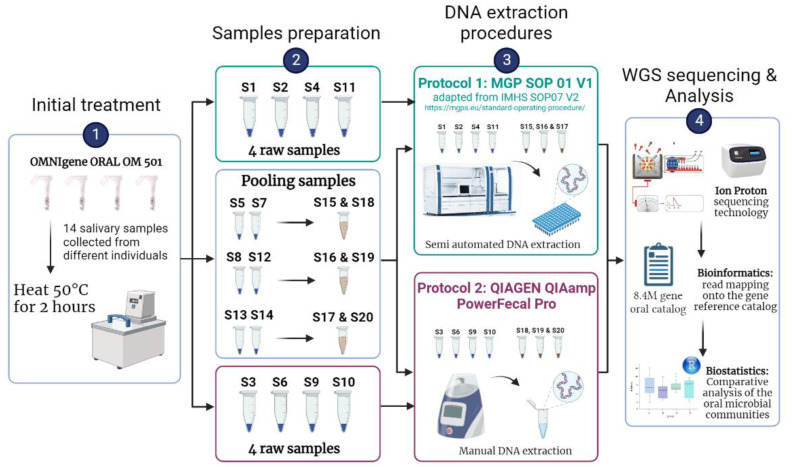
General workflow of the study strategy. Before DNA extraction, samples were pre-treated (step 1) and prepared for DNA extraction procedures, by pooling randomly chosen samples when necessary (step 2). Two different DNA extraction procedures were used: Protocol P1, MGP SOP01 v1 adapted from the IHMS (International Human Microbiome Standards) SOP 07 V2 and Protocol P2 QIAGEN QIAamp PowerFecal Pro (step 3). Finally, samples were sequenced using the WGS Ion Proton sequencing technology before being analyzed using bioinformatical and biostatistical tools (step 4; Methods).

**Figure 2 biomolecules-13-01505-f002:**
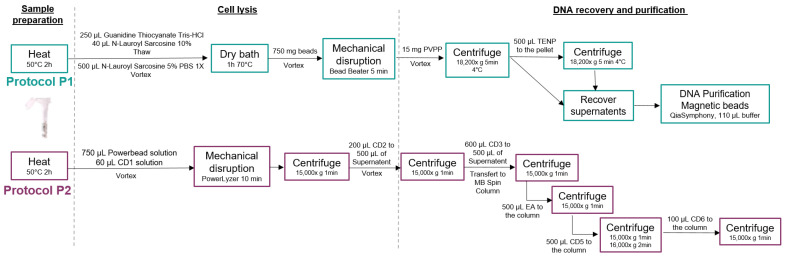
Detailed DNA extraction procedures. Diagrams describing each step for the DNA extraction protocols used in this study. Procotol P1 corresponds to the MGP SOP 01V1 procedure, which is fully available at https://mgps.eu/standard-operating-procedure (accessed on 2 October 2023). Protocol P2 corresponds to the Qiagen QIAamp PowerFecal Pro and is fully available at https://www.qiagen.com (accessed on 2 October 2023).

**Figure 3 biomolecules-13-01505-f003:**
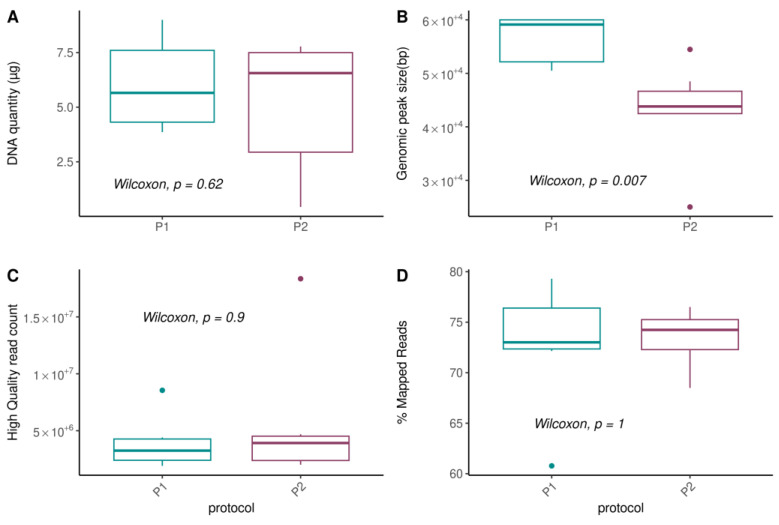
Effects of DNA extraction procedures on technical parameters. (**A**) DNA quantity (µg) obtained using Quant-it fluorescent-based quantification method; (**B**) Size of the genomic peak (bp), an indicator of the DNA quality, obtained using Fragment Analyzer; (**C**) Number of high-quality reads, after removing low-quality and host-related reads; and (**D**) Percentage of reads that mapped onto the 8.4 million oral genes catalog. *p*-values from unpaired Wilcoxon tests are reported.

**Figure 4 biomolecules-13-01505-f004:**
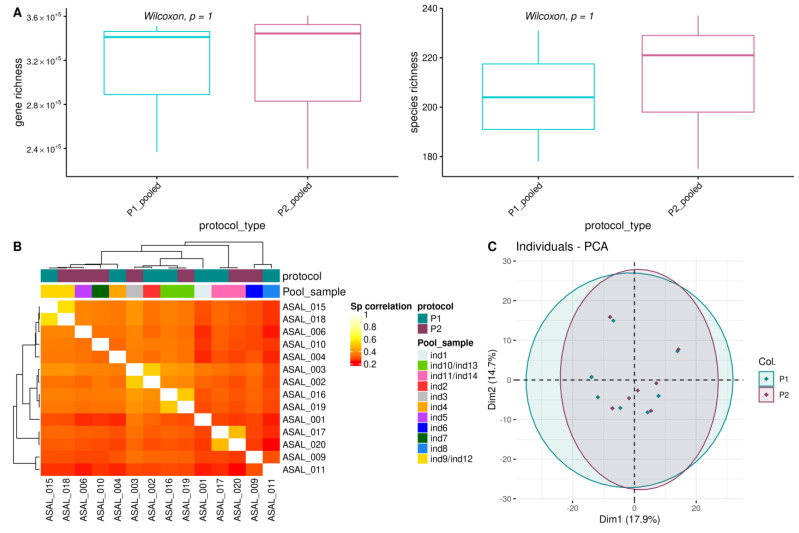
Effects of DNA extraction procedures on the oral microbiome richness and structure. (**A**) Impact on alpha-diversity metrics measured using gene and species richness per protocol on the pooled samples only. *p*-values from unpaired Wilcoxon test are reported; (**B**) Heatmap for Spearman correlations between all pairs of samples using the gene count table. Colored strips referred to Protocols and IDs for raw and pooled samples (Pool sample), respectively; (**C**) Principal Component Analysis (PCA) for all samples using the MSP species count table and colored by protocol. Black circles indicate pooled samples.

**Figure 5 biomolecules-13-01505-f005:**
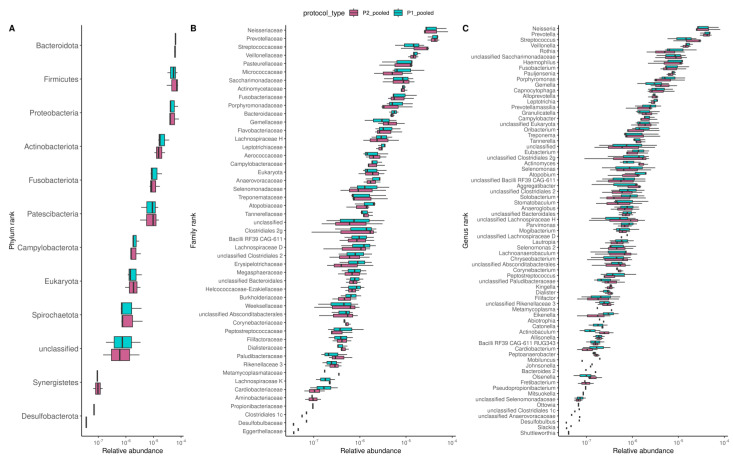
Effects of DNA extraction procedures on the oral microbiome composition. Consistent microbial composition at phylum (**A**), family (**B**), and genus (**C**) taxonomic ranks between samples obtained from P1 and P2 protocols. Boxplots were ordered from the most to the least abundant taxa.

## Data Availability

All sequencing data have been deposited at the European Nucleotide Archive database under the study accession PRJEB59994. The code and associated data to generate the figures are available at https://zenodo.org/record/8288831 (https://doi.org/10.5281/zenodo.8288831).

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
