# Peer review of "Evaluation of an Adapted Semi-Automated DNA Extraction for Human Salivary Shotgun Metagenomics"

_biomolecules, 2023, doi:10.3390/biom13101505_

Round 1
Reviewer 1 Report
Please see the enclosed PDF for e detailed analysis

English is good
Author Response
REVIEWER 1
- The authors should also mention other systemic diseases, such as rheumatoid arthritis and it's treatment. I suggest:
Martu, M.-A.; Maftei, G.-A.; Luchian, I.; Stefanescu, O.M.; Scutariu, M.M.; Solomon, S.M. The Effect of Acknowledged and Novel Anti-Rheumatic Therapies on Periodontal Tissues—A Narrative Review. Pharmaceuticals 2021, 14, 1209. https://doi.org/10.3390/ph14121209
We thank the reviewer for pointing out that the oral microbiome is suspected to play a role in numerous pathologies, beyond Parkinson disease. We have added the suggested reference and also included additional papers that have previously highlighted the relevance of oral microbiota in several diseases.
- Figure B is unclear, please make it larger
We have modified Figure 3 (now Figure 4) to increase its readability (below and in the new version of the manuscript – line 278).
- The discussions section is too short, please document it with more papers from the literature.
We have enhanced the discussion section and added new references that offer new perspectives in oral microbial analysis.
- Please add a limitations of the study section
We have added a new section in the discussion to highlight the limitations of automation and saliva sampling in the study of oral microbiome (lines 358-366).

Reviewer 2 Report
Please see the below comments:
1. The authors presented two protocols P1 and P2. Not any differences of multiple evalutions between P1 and P2 were observed with one exception of genomic peak size. P1 showed significantly higher genomic peak size compared with that from P2. This may be due to different purification strategies (magnetic beads and silica membrane for P1 and P2 protocols, respectively). It means purification strategy based on magnetic beads may result in higher genomic peak size. Semi-automated treatment is not associated with higher genomic peak size.
2. To increase the availability of P1 protocol, the cost and efficiency for DNA purification were needed to be further compared between P1 and P2.
3. If this method was only used to oral microbiome research, it will limit the availability, The authors need to discuss those defects.
Author Response
REVIEWER 2
The manuscript by Meslier et al is very welcome and of great interest, especially in the field of the oral microbiome. The report describes a new semi-automatic method of DNA extraction from saliva, a rich source of microbial DNA. At the same time, the method has the advantage of using saliva, a source of DNA whose collection is less invasive compared to other sources (i.e. gingival). The text is well written overall and clearly understandable therefore, I consider that the article can be published as such. I have only a few recommendations and minor corrections:
We thank the reviewer for his/her general enthusiasm for our work and hope to have satisfied his/her requests.
- “The saliva is nonetheless a complex ecosystem to study, as it is the second most complex microbial ecosystem after fecal microbiota [5], with diverse microorganisms detected such as bacteria, fungus and viruses, hidden behind the human DNA that could represent up to 90% of the DNA sequenced when performing shotgun meta-genomic sequencing [6]” - Could it be reworded? Maybe in two shorter phrases/sentences?
We have rephrased this into two sentences to emphasize better the challenges of studying the oral microbiome in relation to the high amount of human DNA (lines 41-44).
- “2.4. DNA extraction procedures” – I think that the illustration of the two protocols in the form of diagrams would be more useful and facilitate their visualization.
We added a new figure 2 (line 125) providing more details on the protocols in the form of a diagram to facilitate their visualization, as suggested by the reviewer, and consequently renamed and rearranged the other figures.
- Reread the text carefully, there are small spelling mistakes, such as missing or extra spaces.
We thank the reviewer for his/her advice and carefully revised accordingly.
Reviewer 3 Report
The manuscript by Meslier et al is very welcome and of great interest, especially in the field of the oral microbiome. The report describes a new semi-automatic method of DNA extraction from saliva, a rich source of microbial DNA. At the same time, the method has the advantage of using saliva, a source of DNA whose collection is less invasive compared to other sources (i.e. gingival). The text is well written overall and clearly understandable therefore, I consider that the article can be published as such. I have only a few recommendations and minor corrections:
1. “The saliva is nonetheless a complex ecosystem to study, as it is the second most complex microbial ecosystem after fecal microbiota [5], with diverse microorganisms detected such as bacteria, fungus and viruses, hidden behind the human DNA that could represent up to 90% of the DNA sequenced when performing shotgun meta-genomic sequencing [6]” - Could it be reworded? Maybe in two shorter phrases/sentences?
2. “2.4. DNA extraction procedures” – I think that the illustration of the two protocols in the form of diagrams would be more useful and facilitate their visualization.
3. Reread the text carefully, there are small spelling mistakes, such as missing or extra spaces.
Sincerely,
the reviewer
Author Response
REVIEWER 3
- The authors presented two protocols P1 and P2. Not any differences of multiple evalutions between P1 and P2 were observed with one exception of genomic peak size. P1 showed significantly higher genomic peak size compared with that from P2. This may be due to different purification strategies (magnetic beads and silica membrane for P1 and P2 protocols, respectively). It means purification strategy based on magnetic beads may result in higher genomic peak size. Semi-automated treatment is not associated with higher genomic peak size.
We thank the reviewer for this important remark. The protocol P1 presented here shares the lineage of the International Human Microbiome Standards (IHMS) protocols, which was carefully selected among various kits and manual protocols for reliability, reproducibility and quality of the obtained DNA (https://human-microbiome.org/index.php). We therefore expect to produce higher genomic peak size DNA if the automation is as efficient as a manual procedure. We nonetheless agree that the semi-automated treatment is not solely responsible for higher genomic peak size and have clarified this statement in the discussion section.
- To increase the availability of P1 protocol, the cost and efficiency for DNA purification were needed to be further compared between P1 and P2.
The cost and efficiency of DNA purification using automated procedures is indeed an important consideration. We have included this remark in the discussion (lines 331-333).
- If this method was only used to oral microbiome research, it will limit the availability, The authors need to discuss those defects.
As stated in the paper, the P1 protocol suggested here is an adaptation of MGP SOP 01, which is a semi-automated version of the IHMS SOP 07 (also known as “IHMS protocol H”) which was initially developed for stool samples. This protocol is itself based on work from Godon, 1997, to isolate bacterial DNA from fluidized-bed reactor fed by vinasses. We routinely use this protocol to extract DNA from stool and saliva samples and are also evaluating other matrices. We are confident that this protocol can be used to isolate high-quality DNA from a large variety of samples. The discussion was amended to include this (lines 345-348).
Round 2
Reviewer 1 Report
The manuscript has been improved